# Tandem LC-MS Identification of Antitubercular Compounds in Zones of Growth Inhibition Produced by South African Filamentous Actinobacteria

**DOI:** 10.3390/molecules28114276

**Published:** 2023-05-23

**Authors:** Daniel J. Watson, Lubbe Wiesner, Tlhalefo Matimela, Denzil Beukes, Paul R. Meyers

**Affiliations:** 1Division of Clinical Pharmacology, Department of Medicine, University of Cape Town, Cape Town 7700, South Africa; lubbe.wiesner@uct.ac.za (L.W.); mtmtlh001@myuct.ac.za (T.M.); 2School of Pharmacy, University of the Western Cape, Bellville 7535, South Africa; dbeukes@uwc.ac.za; 3Department of Molecular and Cell Biology, University of Cape Town, Cape Town 7700, South Africa; paul.meyers@uct.ac.za

**Keywords:** antitubercular, filamentous actinobacteria, HPLC-MS/MS, natural products, zone of inhibition

## Abstract

Novel antitubercular compounds are urgently needed to combat drug-resistant *Mycobacterium tuberculosis* (Mtb). Filamentous actinobacteria have historically been an excellent source of antitubercular drugs. Despite this, drug discovery from these microorganisms has fallen out of favour due to the continual rediscovery of known compounds. To increase the chance of discovering novel antibiotics, biodiverse and rare strains should be prioritised. Subsequently, active samples need to be dereplicated as early as possible to focus efforts on truly novel compounds. In this study, 42 South African filamentous actinobacteria were screened for antimycobacterial activity using the agar overlay method against the Mtb indicator *Mycolicibacterium aurum* under six different nutrient growth conditions. Known compounds were subsequently identified through extraction and high-resolution mass spectrometric analysis of the zones of growth inhibition produced by active strains. This allowed the dereplication of 15 hits from six strains that were found to be producing puromycin, actinomycin D and valinomycin. The remaining active strains were grown in liquid cultures, extracted and submitted for screening against Mtb *in vitro*. *Actinomadura napierensis* B60^T^ was the most active sample and was selected for bioassay-guided purification. This resulted in the identification of tetromadurin, a known compound, but which we show for the first time to have potent antitubercular activity, with the MIC_90_s within the range of 73.7–151.6 nM against *M. tuberculosis* H37Rv^T^ *in vitro* under different test conditions. This shows that South African actinobacteria are a good source of novel antitubercular compounds and warrant further screening. It is also revealed that active hits can be dereplicated by HPLC-MS/MS analysis of the zones of growth inhibition produced by the agar overlay technique.

## 1. Introduction

Novel antitubercular compounds are urgently needed to combat *Mycobacterium tuberculosis*, the causative agent of tuberculosis (TB). Despite an array of treatment options, TB was estimated to be responsible for 1.6 million deaths in 2021, representing the first increase in deaths since 2005 [1]. While the COVID-19 pandemic is partially responsible, a major cause of this increase is the rapid spread of drug-resistant strains, which are neutralising our first-line treatments. The number of drug-resistant cases of TB is on the rise, with an increase of 450,000 cases of rifampicin-resistant infections observed between 2020 and 2021. The second-line treatments, used to treat drug-resistant cases, see patient non-adherence to the drug regimen due to the high pill burden, long treatment times and toxic side effects, which further complicates treatment [1,2]. This necessitates the discovery and development of improved chemotherapeutics to treat drug-resistant *M. tuberculosis* strains. Furthermore, new treatments should ideally be more efficient (shorter treatment times) and have fewer side effects.

Natural products from the filamentous actinobacteria (actinomycetes) have historically been a rich source of novel antitubercular compounds. Drugs such as streptomycin, the rifamycins, kanamycin and capreomycin were all isolated from filamentous actinobacteria and developed as clinical antitubercular drugs [3,4,5]. However, natural product drug discovery is fraught with difficulties, such as long and expensive studies, which tend to result in the re-isolation of known compounds and have low new molecule hit rates. While these challenges are arduous and intricate, they are not insurmountable and actinobacterial natural products are still a good source of drugs. The advent of new technology and new knowledge regarding the filamentous actinobacteria allow us to develop improved workflows to overcome these challenges and screen these microorganisms for novel antitubercular molecules.

One of the key components in improving natural product screening programs is to dereplicate active samples as soon as possible to prevent the re-isolation of known compounds [6]. This can be achieved using tools such as the Global Natural Product Social (GNPS) molecular networking platform, which has been used globally to identify known and novel compounds [7]. Another crucial aspect is strain selection. Multiple studies have revealed that genetically diverse strains should be targeted as they are more likely to have greater chemical diversity, which increases the likelihood of identifying novel antimicrobial compounds [8,9,10]. These strains include microorganisms isolated from biodiverse environments or that represent rare genera that are difficult to isolate under laboratory conditions [11].

South Africa is the third most biodiverse country in the world, with unique biomes such as the fynbos region [12]. This biodiversity is reflected in the microorganisms isolated from these environments [13,14]. However, few compounds have ever been identified or elucidated from South African filamentous actinobacteria, despite their potential, with most of the literature focusing on the isolation of South African actinobacteria and descriptions of crude extract screening. There are a small number of examples demonstrating the biosynthetic ability of these strains, such as platensimycin and natamycin (pimaricin). However, more work needs to be done on these strains [15,16,17]. Therefore, the aim of this project was to screen South African filamentous actinobacteria for novel antitubercular compounds and isolate the compounds from the most active strains.

## 2. Results and Discussion

### 2.1. Agar Overlay Screening against Mycolicibacterium aurum

The agar overlay technique was used for primary antimycobacterial screening, as it is a cheap and fast semi-quantitative method that allows for the screening of antibacterial activity under a wide range of nutrient growth conditions [18]. Activity is recorded when a zone of growth inhibition is seen after a bacterial indicator is grown over a stab-inoculated filamentous actinobacterium.

Forty-two environmental strains of filamentous actinobacteria from eight genera, namely *Actinomadura, Amycolatopsis, Kribbella, Microbispora, Micromonospora, Nocardia, Nonomuraea* and *Streptomyces*, were screened for antimycobacterial activity using the agar overlay assay against *Mycolicibacterium aurum* strain A+. *M. aurum* was used as an indicator for *M. tuberculosis* as they have a comparable drug susceptibility profile, but *M. aurum* is non-pathogenic and has a faster growth rate [19]. Each filamentous actinobacterial strain was grown on six diverse growth media with the aim of inducing the production of a wide range of secondary metabolites [20].

Thirty-one strains showed activity on at least one type of growth medium, with a total of 79 hits in the primary screen (Appendix A). These results were confirmed by stab-inoculating each active strain individually in the same growth media as those on which activity was recorded and repeating the overlay assays.

Previous studies have shown that the zones of growth inhibition can provide a window into the compounds produced by an active strain. For example, Boya P. et al. (2017) identified several actinomycins and macrolides at the site of interaction on agar between two *Streptomyces* strains and the pathogenic fungus *Escovopsis* TZ49 using MALDI imaging mass spectrometry [21]. Therefore, any antimycobacterial compound detected here would likely be responsible for the activity observed. To analyse these zones, their diameters were measured, and the agar within each zone was excised and extracted before analysis by high-pressure liquid chromatography coupled to a quadrupole time of flight instrument (HPLC-qTOF). The raw spectra were then processed using MSConvert and GNPS molecular networking to identify known antibiotics [7,22]. These hits were subsequently confirmed by manual annotation using MZMine 3 to build chromatograms and PubChem and Kyoto Encyclopedia of Genes and Genomes (KEGG) database searches to support the GNPS identification. This workflow is summarised in Figure 1.

These additional steps allowed the identification and elimination of 15 hits from six different strains, which were found to be known antimycobacterial compounds. These were puromycin, actinomycin D and valinomycin; example chromatograms and annotated product ion spectra of each compound obtained from agar extracts are shown in Figure 2, Figure 3, Figure 4, Figure 5, Figure 6 and Figure 7). These antibiotics are commonly produced by the filamentous actinobacteria and are known to have antimycobacterial activity [23,24,25]. Interestingly, *Kribbella* strain 1B8Tu was found to produce actinomycin D. While actinomycin D is commonly produced by members of the *Streptomyces* genus [4], it has never been reported in members of the genus *Kribbella*. 

To investigate whether the antibiotic production observed in the zones of inhibition matched that in liquid cultures, the six strains producing known compounds were grown in small-scale (25 mL) liquid cultures in their respective media for antibiotic production, extracted and analysed in a similar manner to the zones of inhibition. There was a strong correlation between the molecules identified in the traditional liquid culture extraction method and the zone of inhibition method (Appendix A). This suggested that the analysis of the zones of inhibition from agar plates was a suitable method for the rapid identification of strains producing known compounds. Importantly, strains for which the active compounds could not be identified through analysis of their zones of inhibition could possibly be producing novel antimycobacterial compounds, which warrant further study against *M. tuberculosis*.

### 2.2. Mycobacterium tuberculosis In Vitro Screening

Strains for which the active compound could not be identified through HPLC-MS/MS analysis of their zones of inhibition were grown in small-scale liquid cultures, extracted and submitted for testing against *M. tuberculosis* H37Rv^T^ *in vitro*.

Thirty-four samples showed antitubercular activity *in vitro* in at least one of the growth media (Appendix A). Only samples that showed minimum concentrations required to inhibit the growth of 90% of the *M. tuberculosis* H37Rv^T^ culture (MIC_90_) < 20 µg/mL in at least two growth media were selected for further screening. Eight samples (from seven actinobacterial strains) met these conditions and were retested to confirm the *in vitro* antitubercular activity and screened for *in vitro* cytotoxicity against the Chinese Hamster Ovary (CHO) cell line to determine selectivity (Table 1).

*Streptomyces fractus* MV32^T^ and *Streptomyces* strains CW5 and Y10 all exhibited potent cytotoxicity against the Chinese Hamster Ovary cell line, excluding them from further study [26].

*Streptomyces africanus* CPJVR-H^T^, *Streptomyces* strain HMC5 and *Actinomadura napierensis* B60^T^ all displayed no cytotoxicity up to 100 µg/mL. *A*. *napierensis* B60^T^ was the most active strain, with low microgram/mL activity in all three *M. tuberculosis* test media conditions. Therefore, it was selected for bioassay-guided fractionation studies to identify the active compound that it was producing.

### 2.3. Purification and Identification of Tetromadurin

Eight litres of *A. napierensis* B60^T^ were cultured in ISP-2 and the cell mass and broth were extracted with ethyl acetate and methanol. The crude extracts were pooled and separated by reverse-phase solid-phase extraction on C18 cartridges before fractionation by high-pressure liquid chromatography coupled to a diode array detector (HPLC-DAD). One active fraction was identified and analysed by HPLC-qTOF to identify a major peak with a *m*/*z* of 778.4747.

Analysis of the mass spectrum of the peak at *m*/*z* 778.4747 revealed it to be an ammonium adduct [M + NH_4_^+^] with a mass of 760.4403 Da. Mass searches using Antibase 2017 [27] matched to A-80577 (tetromadurin), a linear tetronate. This identification was corroborated by comparison of the published ultraviolet absorption spectrum and NMR spectra, which showed identical results (Appendix A and Appendix A) [28,29]. The same feature (*m*/*z* 778.4738, [M + NH_4_^+^]) was detected in the agar extract from *A. napierensis*’ zone of growth inhibition during initial screening, providing additional evidence to corroborate its identification as the active antimycobacterial compound. The chromatogram and accurate masses of both tetromadurin features are displayed in Figure 8 and Table 2. The mass spectrum and annotated product ion spectrum of tetromadurin are displayed in Figure 9 and Figure 10.

Tetromadurin is a type I polyketide polyether tetronate originally isolated from *Actinomadura verrucosospora* strain SF2487 (NRRL B-18236) [28,29]. Tetronates are compounds consisting of a linear fatty acid or polyketide chain and a tetronic acid ring [30]. Tetromadurin is structurally related to tetronasin and tetronomycin, all of which are characterised by possessing a cyclohexane ring, a tetrahydropyran ring and at least one tetrahydrofuran ring [28]. Tetromadurin is an ionophore that acts by disrupting chemical ion gradients, which inhibits primary cellular functions [31,32]. As electrochemical gradients are conserved in all organisms, ionophores tend to display a wide range of antimicrobial activity [24,33,34]. Tetromadurin is no exception and possesses antibacterial activity against Gram-positive bacteria, antiplasmodial activity and antiretroviral activity [28,29,32]. As with many other ionophores, it has no effect against Gram-negative bacteria [24,29]. To our knowledge, tetromadurin has never been screened for antitubercular activity. Screening against *M. tuberculosis* H37Rv^T^ *in vitro* revealed that tetromadurin has potent antitubercular activity, with MIC_90_s ranging from 73.7 to 151.6 nM (Table 3) under the different growth conditions. However, when screened for cytotoxicity, it displayed an IC_50_ of 1.94 µM against the CHO cell line. The low selectivity indices suggest that tetromadurin is not selective enough for further study [26]. This is also commonly seen with ionophores, as few make suitable chemotherapeutics due to their broad activity profiles and indiscriminate mechanisms of action [31].

## 3. Conclusions

In conclusion, we were able to demonstrate that the agar overlay method combined with HPLC-MS/MS analysis is an effective tool to rapidly screen filamentous actinobacteria under different nutrient conditions. With a few additional steps, the zones of inhibition produced by active strains can be used to identify known compounds, streamlining the discovery process and prioritising producers of novel compounds. In the literature, the isolation of novel antitubercular compounds from filamentous actinobacteria is usually done by traditional bioassay-guided fractionation and elucidation by high-resolution mass spectrometry and NMR analysis. For example, dumulmycin was recently isolated from a river sediment *Streptomyces* strain and was shown to have strong antitubercular activity *in vitro*, with an MIC_50_ of 27.1 µM [35]. However, few studies have utilised zones of growth inhibition to identify known compounds, and, to our knowledge, none have done so for antimycobacterial compounds or to the extent discussed in this investigation [21,36]. This method is limited by the databases used for dereplication and the size of the zone of growth inhibition. Smaller zones of inhibition may contain known antibacterial agents, which could go undetected if insufficient material is extracted. Excitingly, tetromadurin has been shown to be a novel antitubercular compound, and while *in vitro* screens suggest that it is not suitable for further study, it does represent a unique scaffold for derivation and medicinal chemistry studies. Ionophores show potential as novel chemotherapeutics; if they can be altered or dosed with other molecules to be more selective, they may represent a new frontier in drug development [31,37].

Finally, this study provides further evidence that South African filamentous actinobacteria are sources of potent antibacterial compounds. Further study should focus on additional screening programs and the isolation and identification of the active compounds in the other top strains from this investigation, namely *S. africanus* CPJVR-H^T^ and *Streptomyces* strain HMC5.

## 4. Materials and Methods

### 4.1. Agar Overlay Assay and Zone of Inhibition Extraction

All filamentous actinobacteria were obtained from the Meyers Culture Collection in the Department of Molecular and Cell Biology, Faculty of Science, University of Cape Town, South Africa. These strains were isolated from a variety of terrestrial and aquatic environments in South Africa by members of Dr Meyers’ research group.

Primary screens for antimycobacterial activity were undertaken using the agar overlay method [18]. Briefly, each strain was stab-inoculated into agar plates from the spore mass on a streaked plate using a sterile toothpick under aseptic conditions. The stab-inoculated plates (four actinobacterial strains per plate in the initial screening) were incubated for 7 days at 30 °C. Six growth media were used to test each strain:

International *Streptomyces* Project medium number 2 (4 g yeast extract (Merck), 10 g malt extract (Merck), 4 g glucose (Merck), pH 7.3) (ISP-2) [38];

German Collection of Microorganisms and Cell Cultures medium number 553 (10 g glucose, 5 g peptone (Biolab), 5 g yeast extract, 5 g meat extract (Sigma-Aldrich), 0.74 g CaCl_2_ × 2H_2_O (Biolab), pH 7.2 (DSMZ #553) [39];

Modified Bennett’s medium (10 g glycerol (Merck), 2 g Bacto^TM^ Casitone (Becton Dickinson, BD Biosciences, CA, USA), 1 g yeast extract, 1 g meat extract, pH 7.0) [40];

Czapek Solution agar (30 g sucrose, 2 g NaNO_3_ (Merck), 1 g K_2_HPO_4_ (Merck), 0.5 g KCl (Merck), 0.5 g MgSO_4_.7H_2_O (Merck), 0.01 M FeSO_4_ (Merck), pH 7.2) [40];

Japan Collection of Microorganisms medium number 61 (15 g starch, 4 g yeast extract, 0.5 g K_2_HPO_4_, 0.5 g MgSO_4_.7H_2_O, pH 7.4) (JCM #61) [41]; and

Difco^TM^ Middlebrook 7H9 medium (Becton Dickinson) supplemented with 10% glucose [40].

One day prior to overlay, 4–5 loopfuls of *Mycolicibacterium aurum* strain A+ were inoculated into 10 mL of Middlebrook 7H9 supplemented with 0.05% glycerol and 0.02% Tween 80 (Merck) in a sterile universal tube and incubated at 30 °C with shaking at 100 rpm for 24 h. On the day of overlay, the *M. aurum* culture was streaked for single colonies on Middlebrook 7H9 and Gram-stained to ensure purity [42]. The optical density at 600 nm (OD_600_) was measured and the culture volume required for each plate was determined using the following empirical formula: Volume × OD_600_ = 160.

The calculated volume of *M. aurum* culture was then added to tubes of 6 mL 2 × yeast extract tryptone (2YT) [40] sloppy agar (0.7% agar), mixed and gently poured over each stab-inoculated plate to cover the entire plate, but not the actinobacterial colonies. The overlaid agar was allowed to set and the plates were incubated at 37 °C for 3 days. Activity was recorded if a zone of inhibition of the growth of *M. aurum* A+ was observed around an actinobacterial colony. For the initial screening, four strains were inoculated per plate. Any strains that showed activity were then stab-inoculated individually in the centre of an agar plate of the same medium, incubated and subjected to the overlay assay to confirm activity.

The agar within the zone of inhibition was then excised and added to a 50 mL Falcon tube before freezing at −20 °C. Two millilitres of methanol (MeOH) (>99%, Kimix Chemical and Lab Supplies, Cape Town, South Africa) and 40 mL of ethyl acetate (EtOAc) (≥98%, Merck) were added to each frozen tube and the agar pieces were allowed to thaw at room temperature while shaking at 90 rpm. The agar pieces were filtered from the solvent using two coffee filters (size 1 × 4, House of Coffees, Johannesburg, South Africa) and the solvent layer was washed with 40 mL of Millipore water in a separating funnel. Once the layers had separated, the solvent layer was collected and dried down in a fume hood.

### 4.2. Bacterial Cell Cultivation and Extraction

Strains that showed antimycobacterial activity by the overlay assay (and for which the active compound could not be identified) were grown in the growth media in which activity had been detected and extracted. Frozen actinomycete stock cultures (15% *v*/*v* glycerol) were thawed at room temperature and inoculated into 30 mL of liquid medium in a 250 mL Erlenmeyer flask and incubated for 7–10 days at 30 °C with shaking at 100 rpm. Once sufficient cell mass had been produced, the culture was Gram-stained [42] and streaked for single colonies to determine purity. Whole culture extraction was achieved with the addition of 5 mL MeOH and 50 mL EtOAc. The culture and solvent were agitated for 2 h at 90 rpm at room temperature to assist extraction, before filtering the cell mass through two coffee filters and separating the organic and aqueous layers in a separating funnel. The organic layer was collected and dried down. The cell mass and aqueous layer were discarded.

### 4.3. Antitubercular Testing

The minimum concentrations of the crude extracts and tetromadurin required to inhibit 90% of the culture (MIC_90_) were determined using the broth microdilution assay against *Mycobacterium tuberculosis* strain H37Rv^T^, as described by Soares de Melo et al. (2015) [43] and modified by the H3D Tuberculosis Unit at the University of Cape Town, South Africa. Briefly, *M. tuberculosis* H37Rv^T^ was cultured in Difco^TM^ Middlebrook 7H9 medium (Becton Dickinson) supplemented with 0.4 % glucose, Middlebrook albumin–dextrose–catalase (ADC) enrichment (Becton Dickinson) and 0.05% Tween 80 (7H9_ADC_GLU_ TW) [44]; Difco^TM^ Middlebrook 7H9 medium (Becton Dickinson) supplemented with 0.4% glucose, ADC enrichment (Becton Dickinson) and 0.05% Tyloxapol (7H9_ADC_GLU_ TX); and Difco^TM^ Middlebrook 7H9 medium (Becton Dickinson) supplemented with 0.03% Difco Casitone, 0.4% glucose and 0.05% Tyloxapol (7H9_CAS_GLU_ TX ) [45].

The *M. tuberculosis* H37Rv^T^ cultures were grown until an OD_600_ of 0.6–0.7 was reached and then diluted 1:500. Next, 50 µL of the respective medium was added to each well in a 96-well microtitre plate. Test samples were diluted in dimethyl sulfoxide (DMSO) and 50 µL was added to Row 1. Two-fold serial dilutions were performed in 50 µL volumes down the rest of the plate to Row 10. Next, 50 µL of the diluted *M. tuberculosis* culture was added to each well. This generated a concentration test range of 62.5–0.12 µg/mL. Isoniazid and moxifloxacin were used as positive controls for antimycobacterial activity. The microtitre plate was sealed in a secondary container and incubated at 37 °C with 5% CO_2_ and humidification for 7 days. Alamar Blue reagent was added to each well of the assay plate 24 h prior to the assay end, after which incubation of the microtitre plate was continued. Relative fluorescence (excitation 540 nm; emission 590 nm) was measured using a SpectraMax i3x Plate Reader at day 7. The raw fluorescent data (Relative Fluorescent Units) were acquired using a SpectraMax i3x Plate reader (Molecular Devices Corporation 1311 Orleans Drive Sunnyvale, CA 94089, USA). The Softmax^®^ Pro 6, 4-parameter curve fit protocol was used to calculate the MIC_90_s.

### 4.4. Cytotoxicity Testing

Cytotoxicity assays were conducted against the Chinese Hamster Ovary (CHO) cell line (ATCC CCL-61, strain CHO-K1) by the H3D Parasitology Unit at the University of Cape Town, South Africa. CHO cells were cultured in a medium consisting of 10% foetal calf serum (FCS, Celtic Molecular Diagnostics, Mowbray, South Africa), 45% Dulbecco’s Modified Eagle’s Medium (DMEM, Highveld Biologicals, Lyndhurst, South Africa) and 45% HAMSF12 medium (1:1; Sigma, St. Louis, MO, USA). Cells were grown at 37 °C in a humidified atmosphere of 5% CO_2_ and maintained by passage. Cell lines were seeded in 96-well microtitre plates at 104 cells/well in cell medium. Once seeded, cells were incubated at 37 °C for 24 h before the test samples were added. Each sample was tested in triplicate at six concentration points (100, 10, 1, 0.1, 0.01 and 0.001 µg/mL). Emetine was used as the reference standard for cytotoxicity. Cell viability was determined using the colorimetric 3-(4,5-dimethylthiazol-2-yl)-2,5-diphenyl tetrazolium bromide (MTT) assay as described by Mosmann [46]. One hundred microliters of dimethyl sulfoxide (DMSO, Sigma-Aldrich, Johannesburg, South Africa) was added to dissolve the MTT formazan derivatives, and the plates were gently agitated for 2 min to ensure homogenous mixtures. Absorbance was measured at 540 nm using a ModulusTM II Microplate Reader (Turner BioSystems, Inc. Sunnyvale, CA, USA). The data were analysed by nonlinear regression analysis using the GraphPad PRISM version 4.00 program to determine the IC_50_ of each compound.

### 4.5. Large-Scale Cultivation

*Actinomadura napierensis* B60^T^ [47] was cultured from frozen glycerol stocks (15% *v*/*v)* in 15 mL of ISP-2 liquid medium in a 250 mL Erlenmeyer flask. After 3 days of incubation at 30 °C with shaking, this seed culture was used to inoculate 100 mL ISP-2 in a 1 L Erlenmeyer flask. After 3 days of incubation at 30 °C with shaking, this second seed culture was used to inoculate 1 L ISP-2 in a 5 L Erlenmeyer flask. This culture was incubated for 14 days at 30 °C with shaking. The cell mass was filtered from the broth by vacuum filtration through glass wool and extracted with 350 mL of MeOH, followed by repeated extraction with 350 mL of EtOAc. The broth was extracted thrice with 350 mL EtOAc in a separating funnel. The organic layers were pooled and dried down in a fume hood.

Crude extracts were reconstituted in water/MeOH (50%) and fractioned using Phenomenex^®^ StrataTM X-33 µm Reverse Phase C18 SPE cartridges (200 mg, 3 mL) (Separations, Johannesburg, South Africa). A SPEEDISK^®^ 48 manifold was used to wash solvents and dissolved crude extract samples through the C18 cartridges under pressure. The C18 cartridges were equilibrated before use by washing 2 mL MeOH (99.9%, Honeywell, Johannesburg, South Africa) through each cartridge, followed by 2 mL Millipore water under pressure. The crude extracts were washed onto the cartridges, followed by MeOH and water in different concentrations, starting at 50% and increasing by 10% MeOH, with a final wash step of 100% MeOH. The second, third and fourth fractions (60–80%) were found to be active and were collected, pooled and dried under nitrogen at 37 °C.

The active compound of *A. napierensis* B60^T^ was identified following high-pressure liquid chromatography using a Shimadzu 20-A LC system couped to a CBM-40 diode array detector and FRC-10 fraction collector with a Kinetex^®^ C18 column (5 µm, 100 Å, 150 mm × 6 mm). The mobile phases that were used were 1 mM ammonium formate in water as the aqueous mobile phase and MeOH with 0.05% formic acid as the organic mobile phase. The flow rate used was 0.7 mL/min, with a total runtime of 60 min, starting at 30% organic and increasing to 95% over 45 min; this was held for 2 min and then decreased back to 30% organic to equilibrate the column for the next run. The column oven was set to 30 °C and the scanning range of the DAD was 190–400 nm. The peak was collected from 36.5 to 37 min.

### 4.6. HPLC-MS Analysis and Molecular Networking

High-pressure liquid chromatography linked to quadrupole time of flight (HPLC-qTOF) high-resolution mass spectrometry was performed using an AB Sciex^®^ X500R QTOF coupled to an AB Sciex^®^ Exion LC system. Spectral data were obtained using information-dependent acquisition (IDA) at a mass range of 50–2000 Da. All methods, batches and data were processed using OS Sciex^®^ v1.2. The declustering potential was 80 V, the curtain gas (N_2_) was at 25 pounds per square inch (psi), the ion spray voltage was 5500 V, and the source temperature was 450 °C. Ion source gases 1 and 2 were at 45 and 55 psi, respectively. The collision energy was 10 eV for the MS scans and 20–50 eV for MS/MS scans. The IDA intensity threshold was 50 cycles per second. The aqueous mobile phase used was 1 mM ammonium formate in water, and the organic mobile phase was MeOH/0.5% formic acid. The method used a gradient starting at 2% organic and ending at 98% organic phase, with a flow rate of 700 µL/min and a run time of 35 min. A Kinetex^®^ C18 column (5 µm, 100 Å, 150 mm × 6 mm) with a column protector was used. All solvents were sonicated for 10 min before use to remove bubbles.

Raw HRMS data produced by HPLC-QTOF were converted to mzXML format by ProteoWizard tool MSconvert (version 3.0.10051, Vanderbilt University, Nashville, TN, USA) [22]. The converted files were uploaded to the GNPS molecular networking server and analysed by the molecular networking workflow published by Wang et al. (2016). A molecular network was created using the online workflow (https://ccmsucsd.github.io/GNPSDocumentation/) on the GNPS website (http://gnps.ucsd.edu (accessed on 7 November 2019)). The data were filtered by removing all MS/MS fragment ions within +/−17 Da of the precursor *m*/*z*. MS/MS spectra were window-filtered by choosing only the top 5 fragment ions in the +/−50 Da window throughout the spectrum. The precursor ion mass tolerance was set to 2.0 Da with an MS/MS fragment ion tolerance of 0.5 Da. A network was created where edges were filtered to have a cosine score above 0.7 and more than five matched peaks. The cosine score is a mathematical measure of spectral similarity between the query/experimental and GNPS library spectra, with a score of 1 representing identical spectra and a score of 0 representing no similarity.

Further, edges between two nodes were kept in the network if, and only if, each of the nodes appeared in each other’s respective top ten most similar nodes. Finally, the maximum size of a molecular family was set to 100, and the lowest-scoring edges were removed from molecular families until the molecular family size was below this threshold. The spectra in the network were then searched against GNPS’ unique spectral libraries, and the library spectra were filtered in the same manner as the input data. All matches between network and library spectra were required to have a cosine score above 0.7 and at least five matched peaks. The results were visualised using Cytoscape v3.7.2 [48]. To confirm the identity of the antibiotics identified by GNPS molecular networking, the mzXML files were imported to MZMine version 3.2.8, where the masses were detected to the MS1 and MS2 levels, before a chromatogram was built using the ADAP Chromatogram Builder Module [49,50]. Each known active compound was searched for by the mass given in GNPS and then identified using the PubChem and KEGG databases. To compare the agar and liquid tetromadurin sample features in MZMine, the chromatograms were aligned using the join aligner function in MZMine. The default settings were used for all functions. Product ion spectra were manually annotated with assistance from the Sciex^®^ Explorer tool.

### 4.7. NMR Structural Assignment of Tetromadurin (SF2487/A80577)

One- and two-dimensional NMR spectra were obtained on an Avance Bruker III 400 MHz (Bruker, Billerica, MA, USA) using standard pulse sequences. Samples were dissolved and analysed in CDCl_3_ (δH 7.26, δC 77.23) and CD_3_OD (δH 3.31, δC 49.1). Due to the paucity of isolated natural product, a 13C NMR spectrum could not be obtained. However, the majority of the 13C chemical shifts could be estimated from the HSQC and HMBC NMR spectra (Appendix A and Appendix A).

## Figures and Tables

**Figure 1 molecules-28-04276-f001:**
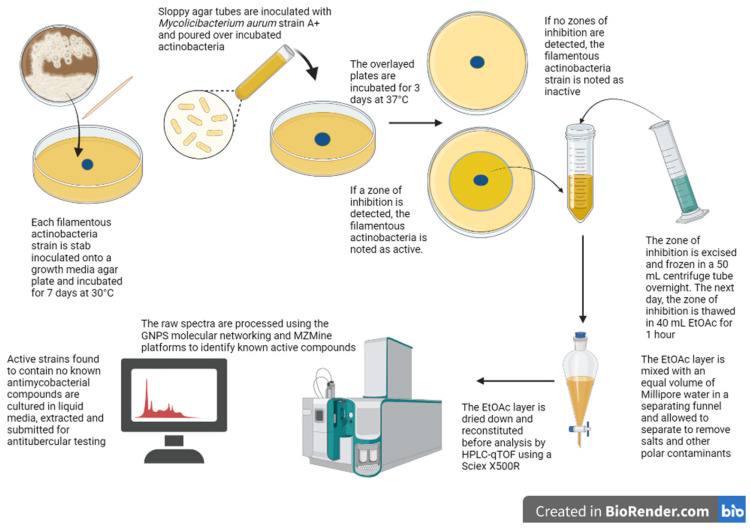
Summary of workflow used to screen filamentous actinobacteria for novel antimycobacterial compounds using the agar overlay assay and HPLC-qTOF analysis to dereplicate active samples. The image was created using Biorender.com.

**Figure 2 molecules-28-04276-f002:**
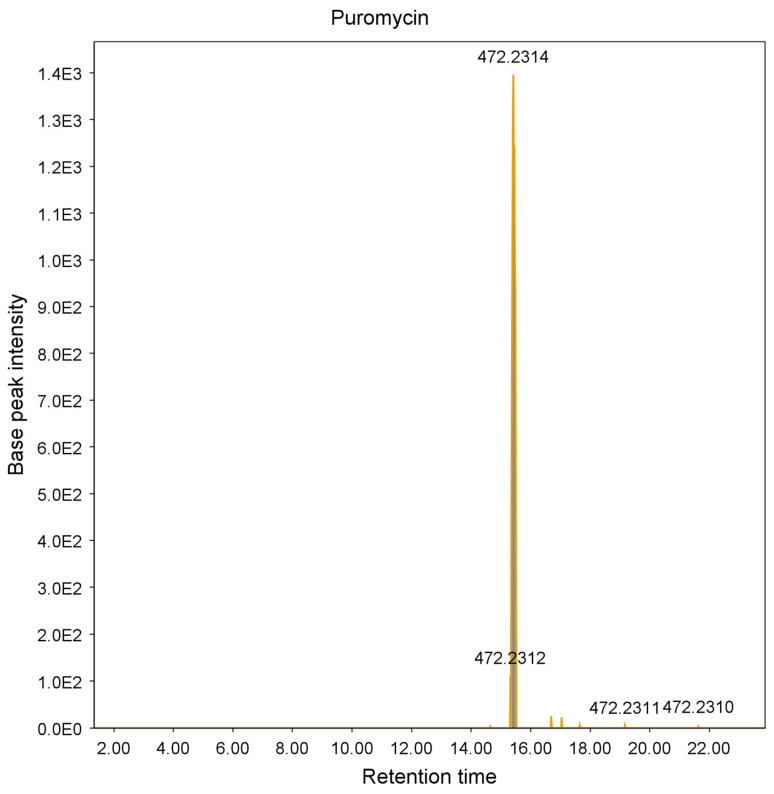
Chromatogram of the puromycin peak [M + H^+^] obtained from the zone of growth inhibition of *M. aurum* strain A+ tested on an ISP 2 plate inoculated with *Streptomyces* strain C23.

**Figure 3 molecules-28-04276-f003:**
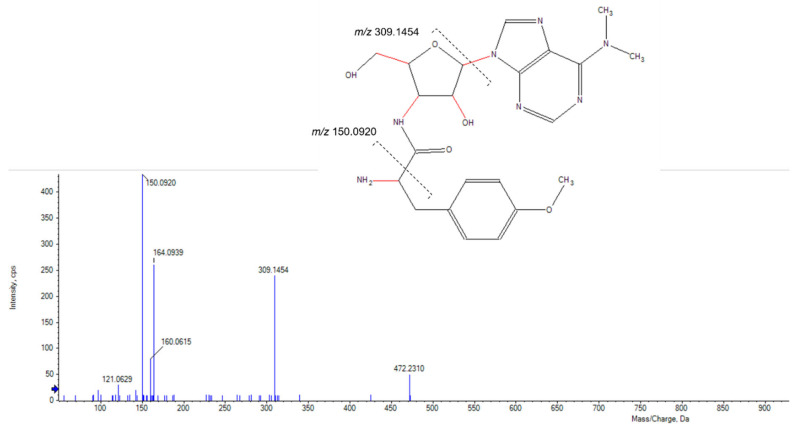
Annotated high-resolution product ion spectrum of puromycin detected in the zone of growth inhibition of *M. aurum* strain A+ tested on an ISP 2 plate inoculated with *Streptomyces* strain C23.

**Figure 4 molecules-28-04276-f004:**
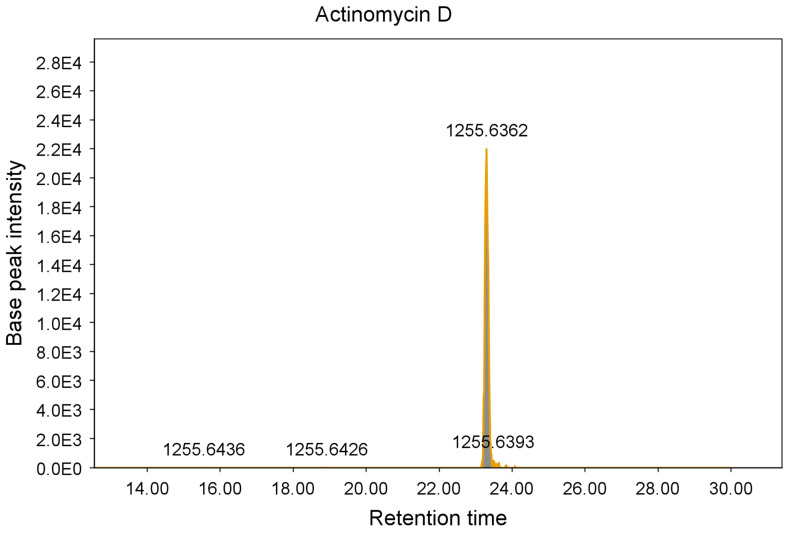
Chromatogram of the actinomycin D peak [M + H^+^] obtained from the zone of growth inhibition of *M. aurum* strain A+ tested on a JCM #61 plate inoculated with *Streptomyces* strain PR10.

**Figure 5 molecules-28-04276-f005:**
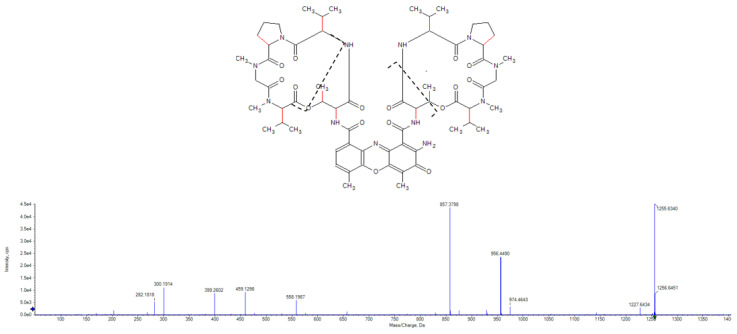
Annotated high-resolution product ion spectrum of actinomycin D detected in the zone of growth inhibition of *M. aurum* strain A+ tested on a JCM #61 plate inoculated with *Streptomyces* strain PR10.

**Figure 6 molecules-28-04276-f006:**
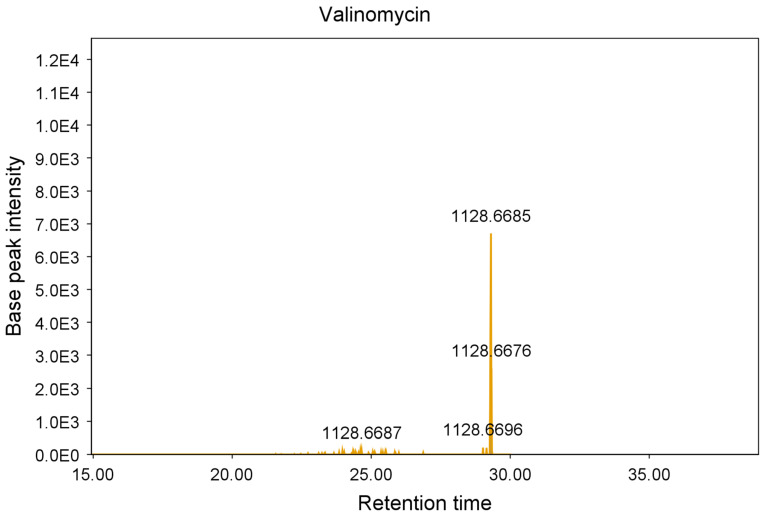
Chromatogram of the valinomycin peak [M + NH_4_^+^] obtained from the zone of growth inhibition of *M. aurum* strain A+ tested on an ISP 2 plate inoculated with *Streptomyces* strain PR3.

**Figure 7 molecules-28-04276-f007:**
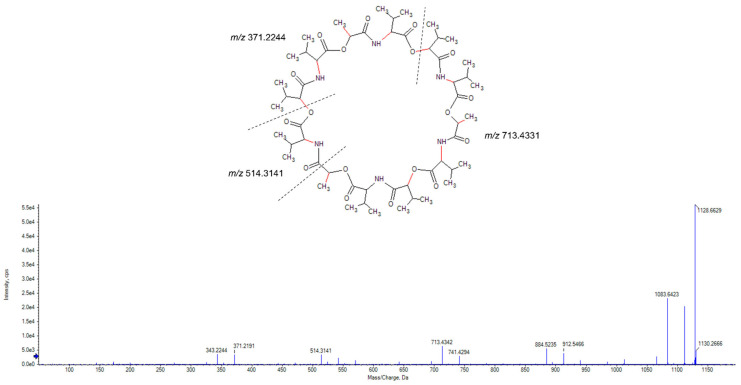
Annotated high-resolution product ion spectrum of valinomycin detected in the zone of growth inhibition of *M. aurum* strain A+ tested on an ISP 2 plate inoculated with *Streptomyces* strain PR3.

**Figure 8 molecules-28-04276-f008:**
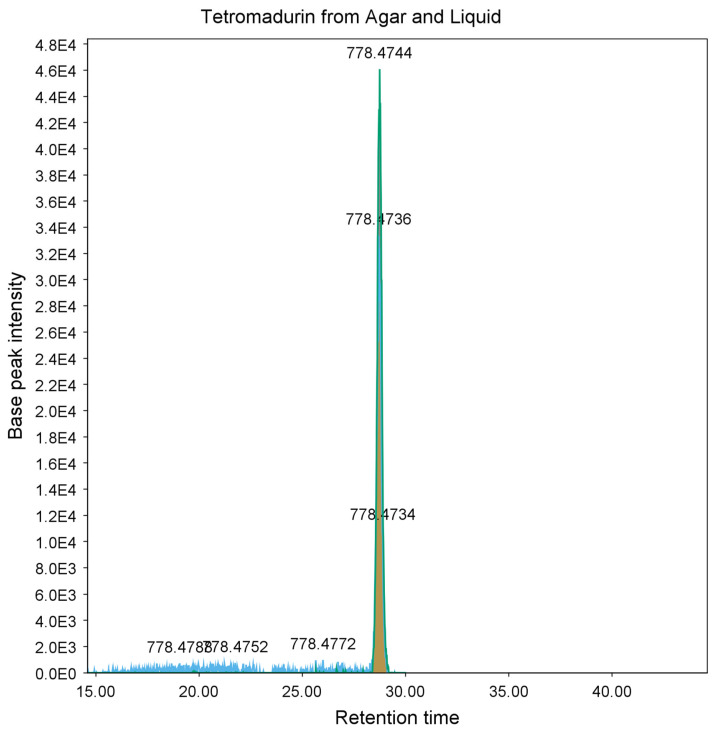
Chromatogram of tetromadurin from agar zone of inhibition (yellow) and from liquid culture following bioassay-guided fractionation (blue).

**Figure 9 molecules-28-04276-f009:**
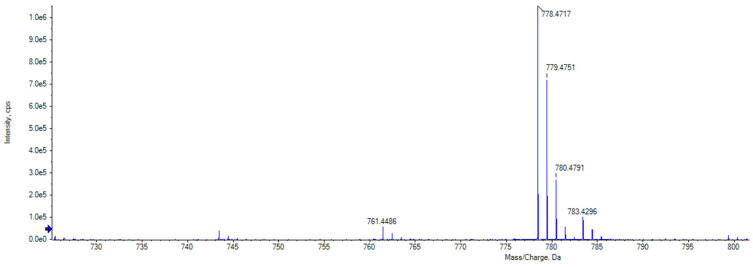
The mass spectrum of tetromadurin identified from liquid cultures of *Actinomadura napierensis* B60^T^ following bioassay-guided fractionation.

**Figure 10 molecules-28-04276-f010:**
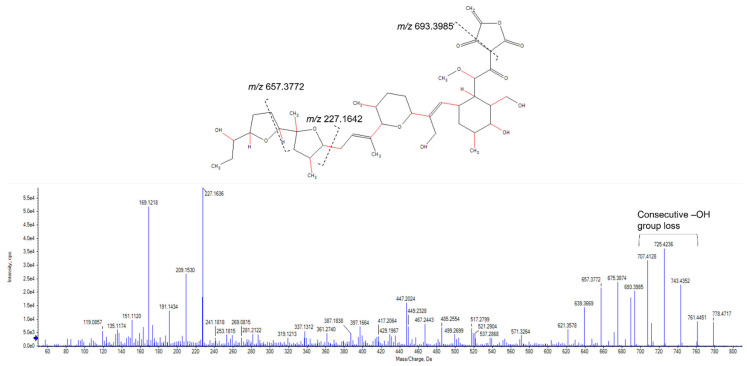
The annotated high-resolution product ion spectrum of tetromadurin from liquid cultures of *Actinomadura napierensis* B60^T^ following bioassay-guided fractionation.

**Table 1 molecules-28-04276-t001:** Mean antitubercular activity *in vitro* and cytotoxicity against the Chinese Hamster Ovary cell line of the most active actinobacterial strains screened. Two technical and biological repeats for antitubercular activity and one biological repeat for cytotoxicity.

Strain Name	Growth Medium	Antitubercular Activity (MIC_90_, µg/mL)			Chinese Hamster Ovary IC_50_ (µg/mL)
		7H9_ADC_GLU_ TX	7H9_CAS_GLU_ TX	7H9_ADC_GLU_ TW	
*Actinomadura napierensis* B60^T^	ISP-2	1.53 ± 0.80	1.25 ± 0.15	1.20 ± 0.80	>100
*Streptomyces africanus* CPJVR-H^T^	ISP-2	8.05 ± 4.6	>62.5 ± ND	0.74 ± 0.35	>100
*Streptomyces* strain CW5	JCM #61	0.34 ± 0.22	0.21 ± 0.036	1.01 ± 0.90	0.032
*Streptomyces* strain HMC5	JCM #61	19.2 ± 6.4	2.75 ± 0.040	14.2 ± 1.3	>100
*Streptomyces fractus* MV32^T^	DSMZ #553	6.30 ± 5.5	<0.12 ± ND	<0.12 ± ND	5.6
*Streptomyces* strain Y10	DSMZ #553	<0.12 ± ND	<0.12 ± ND	<0.12 ± ND	2.4
*Streptomyces* strain Y10	JCM #61	0.59 ± 0.14	<0.12 ± ND	<0.12 ± ND	0.13

**Table 2 molecules-28-04276-t002:** Accurate masses of tetromadurin detected in HPLC-DAD sample from liquid cultures and agar in zone of growth inhibition.

Strain Name	Growth Medium	Source	Compound	Theoretical Monoisotopic Mass (Da)	Experimental Mass (Da) *	Accurate Mass Difference (Da)	Mass Error (ppm)
*Actinomadura napierensis* B60^T^	ISP-2	Agar	Tetromadurin	760.4398	760.4394	0.0004	0.526
		Liquid		760.4398	760.4403	−0.0005	−0.658

* The experimental mass was determined by subtracting 18.0343 Da from the measured mass as both features were detected as ammonium adducts [M + NH4+]. Agar—obtained from zones of growth inhibition; liquid—identified in liquid culture following bioassay-guided fractionation.

**Table 3 molecules-28-04276-t003:** Mean MIC_90_ and IC_50_ of tetromadurin against *M. tuberculosis* H37Rv^T^ and Chinese Hamster Ovary cell line, respectively. Two biological and technical repeats.

Culture Medium	*M. tuberculosis* H37Rv^T^ MIC_90_ (nM)	Chinese Hamster Ovary IC_50_ (µM)	Selectivity Index
7H9_ADC_GLU_TW	148.1 ± 49		13
7H9_ADC_GLU_TX	151.6 ± 70		12
7H9_CAS_GLU_TW	73.7 ± 18	1.94 ± 8.0	26

## Data Availability

MS raw files and GNPS jobs are available at MassIVE MSV000091368. Strains and additional data are available on request.

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
