# Peer review of "Tandem LC-MS Identification of Antitubercular Compounds in Zones of Growth Inhibition Produced by South African Filamentous Actinobacteria"

_molecules, 2023, doi:10.3390/molecules28114276_

Round 1

Reviewer 1 Report

Refer to attached document

Reviewer 2 Report

The authors aimed their work to Screen South African filamentous actinobacteria for novel antitubercular compounds and isolate the compounds from the most active strains. Topic is of much interest. Overall work is interesting and very well written. I have some major comments. If these corrections are made, I believe the above work can be published to Molecules.

  1. A lot of typo and duplicate full stop, Why?
  2. The article not contain line number so I find very hard to add my comments.
  3. Abstract: need to rewrite to show results.
  4. Please kindly ensure that scientific names are italicized.
  5. In table 1: This data would be much better if you included a column for the origins of the isolates, as well as a resistance profile for those isolates.
  6. The NMR spectra aren’t fully described and this can call into question the overall structural elucidation of the obtaining compounds. Please provide copy of NMR spectra in the supporting documents.
  7. Improve the discussions: 

I should have expected the authors to discuss elaborately their findings as compared with other similar studies with antitubercular activity. It is important to build on the discussion to highlight the originality of the work.

  1. Where is statistical analysis?
  2. No conclusion, it is must be separate in section.

Round 2

Reviewer 2 Report

Accept in the present form